# Volatile Compounds Emitted from the Cat Urine Contaminated Carpet before and after Treatment with Marketed Cleaning Products: A Simultaneous Chemical and Sensory Analysis

**Chumki Banik [1], Jacek A. Koziel [1,*] and Elizabeth Flickinger [2]**

[1]  Department of Agricultural and Biosystems Engineering, Iowa State University, Ames, IA 50011, USA; cbanik@iastate.edu
[2]  Kent Pet Group, Muscatine, IA 52761, USA; elizabeth.flickinger@kentww.com
*  Correspondence: koziel@iastate.edu; Tel.: +1-515-294-4206

**Abstract:** Urination on carpet and subflooring can develop into a persistent and challenging problem when trying to mitigate odor. Very little or no information is published on how volatile organic compounds (VOCs) change over time when urine is deposited on a carpet covering a plywood-type subflooring. This research has investigated the VOCs emitted from carpet + subflooring (control), carpet + subflooring sprayed with water (control with moisture), and cat urine-contaminated carpet + subflooring (treatment) over time (day 0 and 15). In addition, the study has recorded the effect of four popular cleaning product applications on VOCs emitted from carpet and evaluated their efficacy in eliminating cat urine related indoor odors over time (days 0 and 15). Carpet-subflooring with all treatments were also contaminated with *Micrococcus luteus*, a nonmotile obligate aerobe commonly found in household dust, to observe the impact of the aerobe on carpet-subflooring VOCs emission. VOCs emitted from carpet + subflooring receiving different treatments were collected from headspace using solid-phase microextraction (SPME). The VOCs were analyzed using a gas chromatography-mass spectrometry olfactometry (GC-MS-O). Many common VOCs were released from the carpet on day 0 and day 15, specifically from urine contamination. Cleaning products were effective in masking several potent odors of cat urine contaminated carpet VOCs on day 0 but were unable to remove the odor that appeared on day 15 in most cases.

**Dataset:** Supplementary material is submitted in the form of a well-organized Excel spreadsheet (Carpet-caturine-cleaningpdt-final).

**Dataset License:** CC-BY

**Keywords:** cat urine; odor mitigation; odor; volatile organic compounds; emission; indoor air quality; solid-phase microextraction; SPME; diffusion; *Micrococcus luteus*

---

## 1. Summary

Research studies have identified several volatile organic compounds (VOCs) in the urine and feces of domestic cat species [1–3]. However, there is no report on volatile compounds from cat urine contaminated carpet material over time. About 25% of U.S. households own a cat as a pet [4], and carpet contributes to approximately 48% of the U.S. total flooring market [5]. In addition to cat urine odor, new carpet materials can also contribute to the indoor VOCs and the health and odor issues generated [6]. Thus, investigation and identification of the types of volatile compounds emitted

from cat urine contaminated carpet material over time are noteworthy. Figure 1 illustrates a few of many ways for humans to be exposed to cat urine or feces when they are deposited on carpet. Bouillard et al. (2005) reported that in office building carpets, bacterial concentrations ranged from $0.73–185 \times 10^5$ CFU·g$^{-1}$, with $7.28 \times 10^5$ CFU·g$^{-1}$ as the median value. *Micrococcus luteus* was among the most commonly isolated microorganisms [7]. Therefore, *M. luteus* was selected to represent microbes commonly found on household surfaces that might interact with cat urine. Solid-phase microextraction (SPME), used to extract the VOCs in this study, is a non-invasive, solventless sampling method to extract and characterize volatiles from a source at trace levels [8–12].

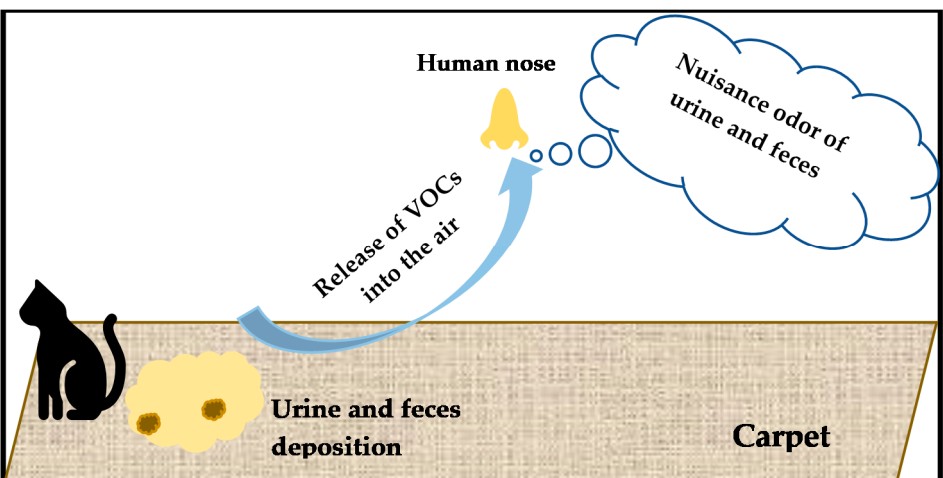

**Figure 1.** Schematic diagram showing the pathway of odorous compounds being emitted from cat excretes deposited carpet to a human nose in a domestic environment.

The objective of this study was to report on VOCs emitted from cat urine contaminated carpet materials, followed by pet odor-controlling cleaning product treatment impacts on those carpet VOCs over time. The analyses were carried out with the assistance of simultaneous chemical and sensory analysis using SPME and multidimensional GC-MS-olfactometry (GC-MS-O). The data presented in the supplementary material can be used to evaluate and assess the indoor air quality in the presence of cat urine contaminated carpet-subflooring over time. The data also contains useful data on the efficacy of four different marketed products that claim to remove cat urine odor from a carpet on the application day and 15 days after application to the carpet.

The key observations are:

1. The cat urine and urine and *M. luteus* treated carpet + subfloorings emitted malodors on day 0 (Table S1). However, the passage of time, aging of the urine, and *M. luteus* treated carpet + subflooring caused the emergence of additional malodors described as strong 'fishy' (most likely trimethylamine) and 'moldy' (most likely 3-octanone).

2. The strong mold-like smell was observed by directly smelling the 15-day aged urine and *M. luteus* treated carpet samples. The 3-octanone's 'mushroom-' or 'mold-like' smell was also detected by using a MS detector for *M. luteus*-urine contaminated test carpet samples (Table S2).

3. On day 0, all four cleaning products treated test carpet + subflooring + urine + *M. luteus* had a profound 'fresh' and 'camphorous' smell when smelled with a human nose, and this observation was supported by the simultaneous chemical sensory analyses via GC-MS-O (Table S3). With the passage of time, the product-treated carpet had a less prominent 'fresh' 'camphorous' smell (Table S4). Also, the 'fish-like' and 'mold-like' odors emerged from the cleaning product-treated carpet + subflooring materials.

## 2. Data Description

The data provided in the Supplementary Material in the Microsoft Excel spreadsheet 'Carpet-caturine-cleaningpdt-final.xlxs,' where the content is organized in sheets (Tables S1–S4) with names as in Table 1. Examples of the Tables S1–S4 content are presented in Tables 2–5.

**Table 1.** Summary of the supplementary tables (Tables S1–S4) and their contents.

| Contents | Table S1 | Table S2 | Table S3 | Table S4 |
|---|---|---|---|---|
| Treatments | VOCs *without* cleaning products | | VOCs *with* cleaning products | |
| Time (day) | 0 | 15 | 0 | 15 |
| Number of volatiles emitted | x | x | x | x |
| Names of the volatiles | x | x | x | x |
| Gas chromatography (GC) column retention time (RT; min) | x | x | x | x |
| MS Spectral Library (NIST & WILEY7) match (%) | x | x | x | x |
| Published odor descriptor | x | x | | |
| Chemical Abstract Service number (CAS#) | x | x | x | x |
| Carpet + sub-flooring | x | x | | |
| Carpet + sub-flooring + water | x | x | | |
| Carpet + sub-flooring + water + *M. luteus* | x | | | |
| Carpet + sub-flooring + cat urine | x | x | | |
| Carpet + sub-flooring + cat urine + *M. luteus* | x | x | | |
| Carpet + sub-flooring + cat urine + *M. luteus* + Product 1 | | | x | x |
| Carpet + sub-flooring + cat urine + *M. luteus* + Product 2 | | | x | x |
| Carpet + sub-flooring + cat urine + *M. luteus* + Product 3 | | | x | x |
| Carpet + sub-flooring + cat urine + *M. luteus* + Product 4 | | | x | x |
| Ion abundance (% relative abundance) | x | x | x | x |

In the 'Appendix A' the Figures A1–A4 present examples of the data collected by sampling headspace gases with DVB/CAR/PDMS Stableflex (2 cm) SPME fiber for 60 min at 37 °C as an overlay of aromagram and total ion chromatogram:

- the carpet + subflooring + urine + microbe on day 0 (Figure A1),
- the Carpet + subflooring + cat urine + microbe on day 15 (Figure A2),
- the Carpet + subflooring + cat urine + microbe + Product 3 on day 0 (Figure A3), and
- the Carpet + subflooring + cat urine + microbe + Product 3 on day 15 (Figure A4).

The simultaneous chemical and sensory analyses (Figures A1–A4) illustrate an important message that intense odorous VOCs (numbered black color peaks in aromagrams) do not necessarily correspond to 'tall' chromatographic peaks (shown in the color red as overlaying chromatograms). This is a well-known phenomenon associated with odor thresholds [13] for odorants that are often in the sub-part-per-billion levels, while odor-causing chemicals are not generating obvious chromatographic signals [14,15].

**Table 2.** Example of a summary of volatiles emitted on day zero from Carpet + Subflooring, carpet + subflooring + water, Carpet + subflooring + water + *M. luteus*, Carpet + subflooring + cat urine, and Carpet + subflooring + cat urine + *M. luteus*. Complete data is summarized in Table S1 (Supplementary Materials, Carpet-caturine-cleaningpdt-final.xlxs).

| # | Compounds | RT (min) | *Library Match % | Odor 'Character' Published/ Recorded | CAS # | Carpet + Subflooring (day 0) | Carpet + Subflooring + Water (day 0) | Carpet + Sub-Flooring + Water + *M. luteus* (day 0) | Carpet + Subflooring + Cat Urine (day 0) | Carpet + Sub-Flooring + Cat Urine + *M. luteus* (day 0) | Ion (% rel. int.) |
|---|---|---|---|---|---|---|---|---|---|---|---|
| | | | | | | MS Detector Response, Peak Area Counts (PACs), Arbitrary Units | | | | | |
| 1 | 2-Pentanone | 5.39 | 74 | | 107-87-9 | | | 178,878 | | | 43(100), 86(35), 71(15), 58(10) |
| 2 | Pentanal | 5.55 | 85 | Fruity, berry | 110-62-3 | 1,353,390 | 2,092,914 | 453,059 | 1,411,250 | 932,775 | 44(100), 41(60), 58(50), 57(40), 43(30) |
| 3 | Octane | 6.74 | 86 | alkane | 111-65-9 | 80,360 | 343,847 | | 287,855 | 245,386 | 43(100), 85(80), 41(70), 57(50), 71(45) |
| 4 | Toluene | 7.11 | 88 | | 108-88-3 | 390,976 | 573,315 | | 678,603 | 249,961 | 91(100), 92(60), 65(10), 51(5) |
| 5 | Hexanal | 8.06 | 95 | Green, leafy, fruity | 66-25-1 | 20,876,330 | 36,530,640 | 12,196,949 | 29,052,691 | 22,302,835 | 56(100), 41(95), 44(90),57(70), 43(60), 39(40) |
| 6 | 1,2-Dimethylcyclopropane | 8.83 | 63 | | 2511-95-7 | 167,106 | | | | 441,161 | 55(100), 70(90), 42(80), 41(50), 39(30), 43(20) |
| 7 | 1-Pentanol | 9.37 | 76 | Sweet, balsam | 71-41-0 | 1,973,782 | 5,537,880 | 3,520,261 | 4,733,304 | 4,656,524 | 55(100), 42(80), 41(75), 70(70), 39(35), 57(30) |
| 8 | 1,2-Dimethylcyclohexane | 9.85 | **50** | | 2207-01-4 | | | | 1,120,465 | 326,959 | 97(100), 41(80), 81(70), 55(70), 70(65), 39(50), 69(40) |
| 9 | 2,4-Octadienal | 10.23 | **54** | Green, fruity | 30361-28-5 | 429,872 | 1,777,135 | | 1,721,952 | 273,430 | 81(100), 69(35), 124(25), 53(20) |
| 10 | 2-Heptanone | 10.54 | 79 | Cheesy, banana, woody | 110-43-0 | 1,714,716 | 3,091,780 | 2,777,763 | 3,923,538 | 2,256,583 | 43(100), 58(75), 71(15), 114(10) |
| 11 | p-Mentha-1,5,8-triene | 10.82 | **42** | roasted | 21195-59-5 | 9,665,111 | 32,947,411 | 38,148,205 | 34,872,658 | 33,253,410 | 119(100), 93(90), 91(70), 92(40), 134(30) |
| 12 | Styrene | 11.24 | 94 | Sweet, balsam, floral, plastic | 100-42-5 | 606,000 | | 389,091 | 602,067 | | 104(100), 103(40), 78(40), 77(20), 51(15) |

Note: * library spectral match <60% are **bolded**.

**Table 3.** Example of a summary of volatiles emitted on day 15 from carpet + subflooring, carpet + subflooring + water, carpet + subflooring + cat urine, and carpet + subflooring + cat urine + *M. luteus.* Complete data is summarized in Table S2 (Supplementary Materials, Carpet-caturine-cleaningpdt-final.xlxs).

| # | Compounds | RT (min) | *Library Match % | Odor 'Character' Recorded/ Published | CAS # | Carpet + Sub-Flooring (day 15) | Carpet + Sub-Flooring + Water + *M. luteus* (day 15) | Carpet + Sub-Flooring + Cat Urine (day 15) | Carpet + Sub-Flooring + Cat Urine + *M. luteus* (day 15) | Ion (% Relative Intensity) |
|---|---|---|---|---|---|---|---|---|---|---|
| | | | | | | MS Detector Response, Peak Area Counts (PACs), Arbitrary Units | | | | |
| 1 | Acetone | 3.19 | 68 | | 67-64-1 | 152,540 | | | 147,139 | 43(100), 58(30), 42(10), 39(5) |
| 2 | Acetic acid methyl ester | 3.33 | 63 | Sweet. fruity | 79-20-9 | | | 1,499,733 | | 43(100), 74(35), 59(15), 42(10) |
| 3 | 2-Pentanone | 5.39 | 72 | | 107-87-9 | 1,704,070 | | | 241,509 | 43(100), 86(35), 71(15), 58(10) |
| 4 | Octane | 6.74 | 75 | | 111-65-9 | 338,270 | 750,351 | | | 43(100), 85(80), 41(70), 57(50), 71(45) |
| 5 | Toluene | 7.11 | 81 | | 108-88-3 | 464,988 | | | 465,549 | 91(100), 92(60), 65(10), 51(5) |
| 6 | 2-Hexanone | 7.81 | 74 | | 591-78-6 | 380,781 | | | | 43(100), 58(60), 100(20), 57(20), 85(15), 71(10) |
| 7 | 1-Pentanol | 9.37 | 76 | Balsamic, sweet | 71-41-0 | | | 3,766,579 | | 55(100), 42(80), 41(75), 70(70), 39(35), 57(30) |
| 8 | 2-Methylpyrazine | 10.03 | 79 | | 109-08-0 | 818,701 | | | 1,053,616 | 64(100), 67(45), 40(25), 53(15), 81(10) |
| 9 | 4-Pyridinamine | 10.3 | 79 | | 504-24-5 | | | 797,924 | 1,145,074 | 94(100), 67(40), 40(35), 53(10) |
| 10 | 2-Heptanone | 10.54 | 79 | | 110-43-0 | 1,882,830 | | 8,674,944 | 2,855,575 | 43(100), 58(75), 71(15), 114(10) |
| 11 | α-Pinene | 10.85 | 91 | | 80-56-8 | 137,164,634 | 119,185,423 | 74,314,748 | 94,769,143 | 93(100), 91(40), 119(30), 77(30), 79(20), 105(12) |
| 12 | Styrene | 11.24 | 91 | Balsamic, plastic | 100-42-5 | 838,213 | 1,463,607 | 1,513,994 | 803,525 | 104(100), 103(40), 78(40), 77(20), 51(15) |

Note: * library spectral match <60% are **bolded**.

**Table 4.** Example of a summary of volatiles emitted on day zero from carpet + subflooring + cat urine + *M. luteus* + Product 1, carpet + subflooring + cat urine + *M. luteus* + Product 2, carpet + subflooring + cat urine + *M. luteus* + Product 3, carpet + subflooring + cat urine + *M. luteus* + Product 4. Complete data is summarized in Table S3 (Supplementary Materials).

| # | Compounds | RT (min) | *Library Match % | CAS # | U + *M.l* + P1 (day 0) | U + *M.l* + P2 (day 0) | U + *M.l* + P3 (day 0) | U + *M.l* + P4 (day 0) | Ion (% Relative Intensity) |
|---|---|---|---|---|---|---|---|---|---|
| | | | | | MS Detector Response, Peak Area Counts (PACs), Arbitrary Units | | | | |
| 1 | Acetone | 3.24 | 68 | 67-64-1 | 358,744 | 380,560 | 474,903 | 430,839 | 43(100), 58(30), 42(10), 39(5) |
| 2 | 2,4-Hexadienal | 5.33 | 81 | 110-62-3 | 131,308 | | | | 81(100), 96(40), 53(30), 65(10) |
| 3 | Pentanal | 5.56 | 88 | 110-62-3 | 1,354,983 | 1,653,133 | 725,418 | 1,945,524 | 44(100), 41(60), 58(50), 57(40), 43(30) |
| 4 | Toluene | 7.11 | 85 | 108-88-3 | 808,174 | 893,198 | | 1,315,643 | 91(100), 92(60), 65(10), 51(5) |
| 5 | Hexanal | 8.06 | 95 | 66-25-1 | 24,237,673 | 29,383,184 | 11,078,995 | 15,770,078 | 56(100), 41(95), 44(90),57(70), 43(60), 39(40) |
| 6 | (Z)-2-Pentene | 8.83 | 63 | 646-04-8 | | | | 1,139,618 | 55(100), 70(90), 42(80), 41(50), 39(30), 43(20) |
| 7 | 1-Pentanol | 9.37 | 72 | 71-41-0 | | 6,742,800 | 6,744,690 | 9,292,407 | 55(100), 42(80), 41(75), 70(70), 39(35), 57(30) |
| 8 | Pentane, 1-chloro- | 9.4 | **59** | 543-59-9 | 4,209,417 | | | | 55(100), 42(90), 41(75), 71(60), 39(35), 57(30) |
| 9 | 2-Butylfurun | 10.21 | 68 | 4466-24-4 | | 1,463,055 | | 1,006,109 | 81(100), 41(30), 55(30), 69(25), 124(25), 53(20) |
| 10 | 2-Heptanone | 10.54 | 86 | 110-43-0 | 2,818,994 | 6,796,498 | 990,983 | 5,096,800 | 43(100), 58(75), 71(15), 114(10) |
| 11 | α-Pinene | 10.8 | 91 | 80-56-8 | 16,649,718 | 46,983,540 | 9,096,994 | 25,095,633 | 93(100), 91(40), 92(35), 119(30), 77(30), 69(25), 41(25) |
| 12 | 1-Butanol, 3-methyl-, acetate | 10.97 | 76 | 123-92-2 | | | 1,560,496 | | 43(100), 70(55), 61(30), 55(30),42(25) |

Note: U + *M.l* + P1 (day 0) = Carpet + sub-flooring + cat urine + *M. luteus* + Product 1 (day 0). U + *M.l* + P2 (day 0) = Carpet + sub-flooring + cat urine + *M. luteus* + Product 2 (day 0). U + *M.l* + P3 (day 0) = Carpet + sub-flooring + cat urine + *M. luteus* + Product 3 (day 0). U + *M.l* + P4 (day 0) = Carpet + subflooring + cat urine + *M. luteus* + Product 4 (day 0). * library spectral match <60% are **bolded.**

**Table 5.** Example of a summary of volatiles emitted on day 15 from carpet + subflooring + cat urine + *M. luteus* + Product 1, carpet + subflooring + cat urine + *M. luteus* + Product 2, carpet + subflooring + cat urine + *M. luteus* + Product 3, carpet + subflooring + cat urine + *M. luteus* + Product 4. Complete data is summarized in Table S4 (Supplementary Materials).

| # | Compounds | RT (min) | *Library Match % | CAS # | U + *M.l* + P1 (day 15) | U + *M.l* + P2 (day 15) | U + *M.l* + P3 (day 15) | U + *M.l* + P4 (day 15) | Ion (% Relative Intensity) |
|---|---|---|---|---|---|---|---|---|---|
| | | | | | MS Detector Response, Peak Area Counts (PACs), Arbitrary Units | | | | |
| 1 | Acetone | 3.19 | 63 | 67-64-1 | | | 643,351 | 332,952 | 43(100), 58(30), 34(5) |
| 2 | Acetic acid methyl ester | 3.34 | 63 | 79-20-9 | | 196,721 | 4,500,806 | | 43(100), 74(30), 59(10) |
| 3 | 2-Pentanone | 5.39 | 79 | 107-87-9 | | | | 1,981,899 | 43(100), 86(20), 58(15), 71(10) |
| 4 | 1-Hexene 4-methyl | 8.52 | 63 | 3769-23-1 | | | 2,200,691 | | 41(100), 57(90), 56(80), 55(70), 70(60), 39(40), 58(20) |
| 5 | 1-Pentanol | 9.4 | 63 | 71-41-0 | | | 2,211,532 | | 55(100), 42(80), 41(75), 70(60), 39(40) |
| 6 | 2-Butylfuran | 10.33 | 85 | 4466-24-4 | | 832,292 | | | 81(100), 124(25), 82(20), 53(10) |
| 7 | Methyl pyrazine | 10.32 | 76 | 109-08-0 | 874,537 | | | 1,435,565 | 94(100), 67(40), 39(15), 53(10) |
| 8 | 2-Heptanone | 10.56 | 83 | 110-43-0 | 1,452,480 | | 6,756,435 | 1,881,383 | 43(100), 58(70), 71(20), 114(10) |
| 9 | α-Pinene | 10.84 | 89 | 80-56-8 | 99,017,729 | 94,473,919 | 61,479,562 | 57,686,178 | 93(100), 119(60), 91(50), 92(40), 77(30), 79(20), 134(20) |
| 10 | Styrene | 11.25 | 95 | 100-42-5 | 1,324,148 | 966,762 | 4,458,613 | 1,322,208 | 104(100), 103(50), 78(45), 77(20), 51(20) |
| 11 | Camphene | 11.42 | 95 | 79-92-5 | 2,001,325 | 1,807,294 | | 1,933,728 | 93(100), 121(80), 91(40), 79(35), 67(30), 136(20) |
| 12 | Verbenene | 11.72 | **53** | 4080-46-0 | 3,218,321 | 6,737,442 | 5,177,868 | 2,561,949 | 91(100), 92(50), 119(20), 105(10), 65(10) |

Note: U + *M.l* + P1 (day 15) = Carpet + sub-flooring + cat urine + *M. luteus* + Product 1 (day 15). U + *M.l* + P2 (day 15) = Carpet + sub-flooring + cat urine + *M. luteus* + Product 2 (day 15). U + *M.l* + P3 (day 15) = Carpet + sub-flooring + cat urine + *M. luteus* + Product 3 (day 15). U + *M.l* + P4 (day 15) = Carpet + sub-flooring + cat urine + *M. luteus* + Product 4 (day 15). * library spectral match <60% are **bolded**.

## 3. Methods

### 3.1. Cat Urine Collection and Sample Preparation

Cat urine was collected from healthy adult group-housed cats at the local Humane Society by substituting nonabsorbent plastic beads for their regular cat litter in a litter box. Urine was allowed to accumulate overnight, then filtered through cheesecloth to remove solid contaminants and stored in a sealed container at 4 °C. Urine was used within 30 d of collection to preserve its original character.

*Micrococcus luteus* ATCC 10240 was aerobically grown in Difco™ Nutrient Broth (BD, Franklin Lakes, NJ, USA) in a rotary shaker water bath at 30 °C for approximately 24 h to an optical density of 2.0–2.5 at 600 nm. Cultures were serially diluted in 3 M Butterfield's phosphate buffer to a final concentration of $10^5$ CFU·mL$^{-1}$. To ensure viability, cultures were used within 4 h of dilution.

The carpet and sub-flooring square (2 × 2 cm) were assembled using a rubber band. The cat urine sample was brought to room temperature before applying it to the test carpet samples. After gentle inverting the urine sample a few times to homogenize it, 2.5 ± 0.5 mL of urine solution was pipetted within a (38 mm) 1.5-inch circle centered on the test carpet specimen using a stainless-steel staining ring. For a control sample, the same procedure was followed to apply de-ionized (DI) water on the carpet instead of cat urine. The carpet specimen was tied with the specimen subflooring using a latex rubberband for all treatments and control used in this study.

*M. luteus* (1 mL) was pipetted on to the appropriate samples. After waiting for 5 min for the samples to soak in, these contaminated carpet samples were put inside a 500 mL wide mouth (9.1 cm outer diameter) glass jar containing 10 mL of DI water and a petri dish separator inside (Figure A5). For the product application, the urine and *M. luteus* contaminated carpet samples were dried overnight, and then the following morning, cleaning products were applied according to package directions. During the incubation period, all jars were always kept at open atmospheric conditions by removing the green sampling septa out from the jar-lid, to have oxygen exchange for the microbes to be alive.

On the sampling day, the green septa were put back on the lid, and the jar was equilibrated to accumulate the VOCs for 1 h before sample extraction. A timer was always used to keep track of the time for each sample. The sampling jar was then put on a digital hot plate set at the desired temperature (37 °C) and a SPME fiber was inserted then and exposed for another hour to complete the VOC extraction from the headspace (Figure 2, Figure A6).

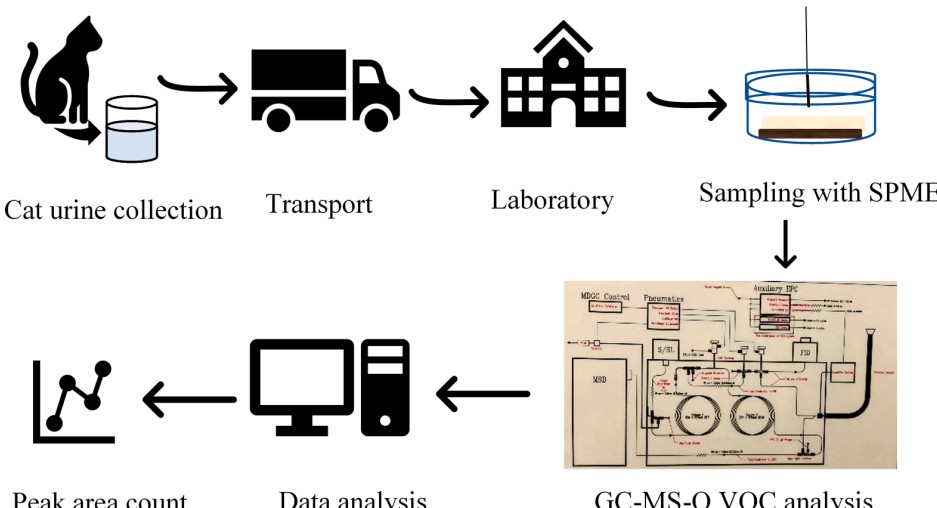

Cat urine collection    Transport    Laboratory    Sampling with SPME

Peak area count    Data analysis    GC-MS-O VOC analysis

**Figure 2.** Schematic of the method for the sample collection to analysis of volatile organics emitted by a test carpet after treatments with water, cat urine, *M. luteus,* and four cleaning products from the market on day 0 and on day 15 were extracted by using 2 cm DVB/PDMS/Carboxen solid-phase microextraction (SPME) fiber. A controlled sample with only carpet and subflooring was also exposed to the SPME fiber similarly as treated carpet material.

*3.2. Data Acquisition and Analysis*

All chemicals emitted from the sample vial headspace were extracted using a SPME fiber of 2 cm 50/30 μm DVB/PDMS/Carboxen (57299-U, Supelco, Bellefonte, PA, USA). All samples were heated to 37 °C (close to body temperature) using a digital hotplate during extraction to enhance the emissions. A 60 min extraction time was used for all of the extractions. After the extraction, the SPME fiber loaded with odorants inserted to the 250 °C GC injector for thermal desorption of samples to the GC columns and separation and analysis using MS and olfactometer.

All carpet, urine, and water treated carpet, urine-microbe and water-microbe treated carpet, and cleaning product treated carpet samples were analyzed and completed using a custom multidimensional gas chromatograph (Microanalytics, Round Rock, TX, USA) built on Standard Agilent 6890N (G1530N) (Agilent Technologies, Santa Clara, CA, USA)-mass spectrometer (Agilent 5973N, Agilent Technologies, Santa Clara, CA, USA) olfactometer (GC-MS-O). The system Automation software was Multitrax v. 10.1 (MOCON, INC. Round Rock, TX, USA), the data acquisition software was Chemstation ver. D.02.00.275 (Agilent Technologies, Santa Clara, CA, USA), and aroma characterization was done using AromaTrax ver. 10.1 (Microanalytics, Round Rock, TX, USA).

The GC oven temperature was programmed at the initial 40 °C for 3 min, followed by ramping up to 240 °C at a rate of 7 °C·min$^{-1}$, where it was maintained for 8.43 min. The quadrupole MS used the electron ionization mode with ionization energy of 70 eV during operation, and the full scan range was 34 to 350 *m/z*. The carrier gas used was ultra-high pure (UHP) helium (99.99%, Airgas, Des Moines, IA, USA). The GC used in this experiment has two columns in a series: the first one is a non-polar column with a fixed restrictor pre-column, and the second one is a polar column. The system was used in full heart cut mode with a total run time of 40 min. One part of the effluent from the polar column was simultaneously directed to the single quadrupole mass spectrometer and three parts of the effluent was directed to the sniff port of the Olfactometry.

The peaks were identified using PBM-Benchtop software (Wiley7 library) and the NIST database library. Aromagrams for odors were generated using AromaTrax software, recorded and generated by the panelist. The odor intensity reported was on a scale of 0–100, where 0 was minimum, and 100 was the maximum. Odors characters recorded/reported by the panelist was verified with published odors descriptors [16,17].

## 4. User Notes

Tables S1–S4 are presented in the Supplemental Material spreadsheet. This spreadsheet has the same layout of columns and column descriptions as Tables 2–5, respectively, in the main text.

**Supplementary Materials:** The following are available online at http://www.mdpi.com/2306-5729/5/4/88/s1, Table S1: contains information of volatile compounds emitted on day 0 data, Table S2: contains information on volatile compounds emitted on day 15 data, Table S3: contains information of volatile compounds emitted from commercial odor removal product applied to urine + *M. luteus* contaminated carpet on day 0 data, Table S4: contains information of volatile compounds emitted from commercial odor removal product applied to urine + *M. luteus* contaminated carpet on day 15 data.

**Author Contributions:** Conceptualization, J.A.K. and C.B.; methodology, C.B. investigation, C.B.; resources, J.A.K. and E.F.; data curation, C.B.; writing—original draft preparation, C.B.; writing—review and editing, J.A.K. and E.F.; visualization, C.B.; supervision, J.A.K. and E.F.; project administration, J.A.K.; funding acquisition, J.A.K. All authors have read and agreed to the published version of the manuscript.

**Funding:** This research was funded by KENT PET GROUP, grant number GR-021366-00001. Iowa Agriculture and Home Economics Experiment Station: project number IOW05556 (Future Challenges in Animal Production Systems: Seeking Solutions through Focused Facilitation, sponsored by Hatch Act and State of Iowa funds; Jacek Koziel).

**Acknowledgments:** The authors would like to thank Baitong Chen, Myeongseong Lee, and Peiyang Li for providing technical help in the laboratory.

**Conflicts of Interest:** The authors declare no conflict of interest. The funders approved the design of the study, cat urine collection, and microbial inoculant preparation. The funders had no contribution to analyses or interpretation of data, to the writing of the manuscript, or to the decision to publish the results.

## Appendix A

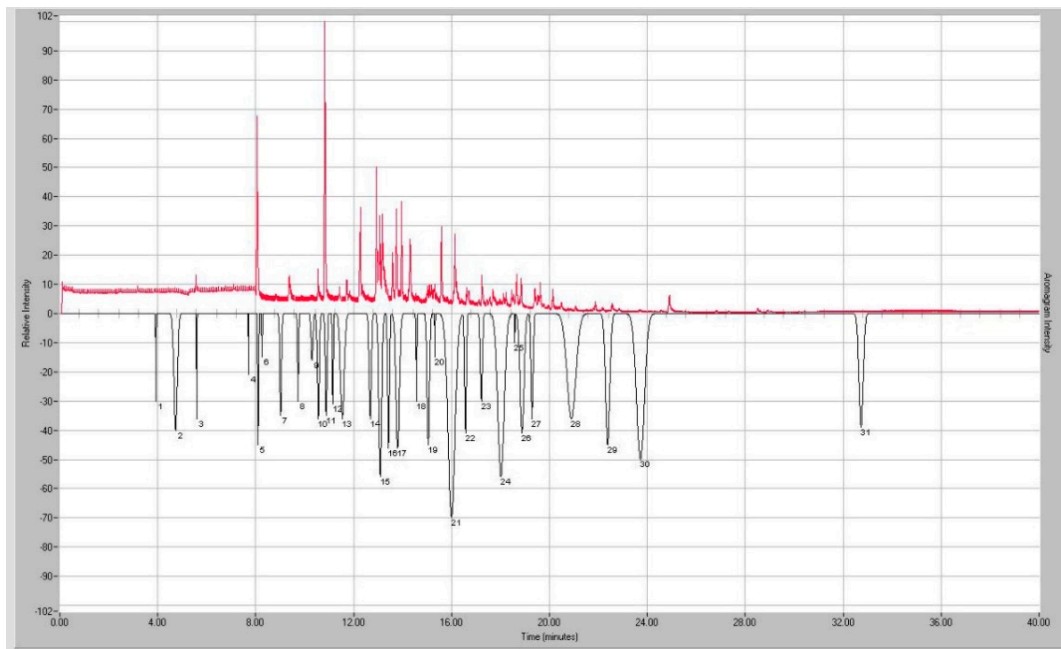

**Figure A1.** An overlay of aromagram (black line) and total ion chromatogram (TIC, red line) of the carpet + subflooring + urine + microbe extracted from headspace by DVB/CAR/PDMS Stableflex (2 cm) SPME fiber for 60 min at 37 °C on day 0. The height of aromagram peaks represents measured odor intensity (percent relative scale). The TIC signal is collected simultaneously, enabling linking odors to specific chemicals. Several unpleasant (out of 31 total) odors were recorded during analysis with GC-MS-O, and few of them were medium-to-strong intensity.

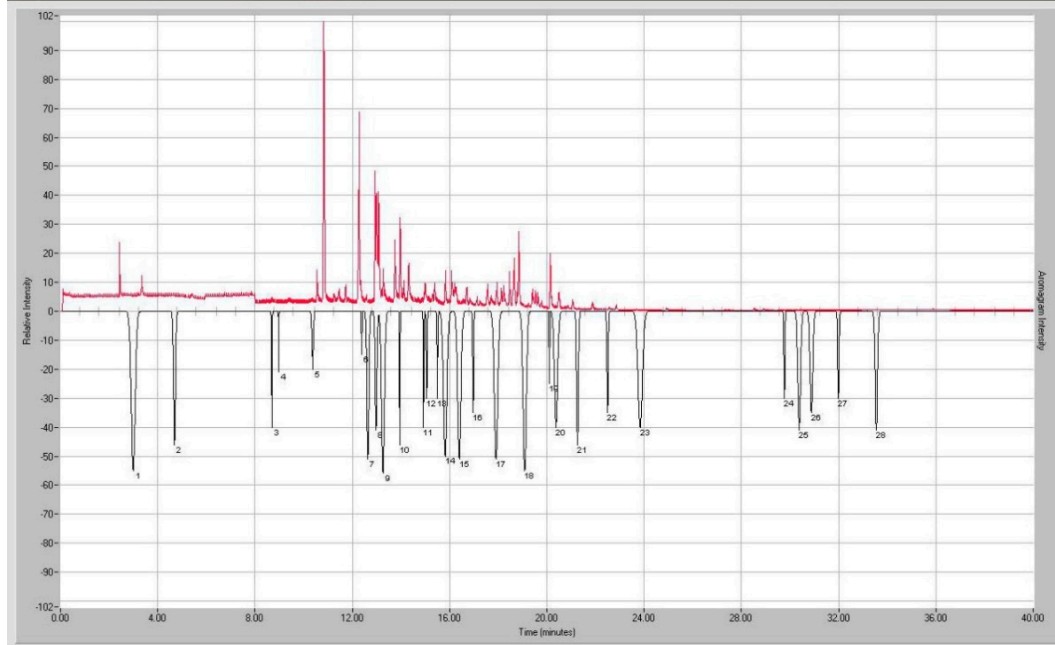

**Figure A2.** An overlay of aromagram (black line) and total ion chromatogram (TIC, red line) of the carpet + subflooring + urine + microbe extracted from headspace by DVB/CAR/PDMS Stableflex (2 cm) SPME fiber for 60 min at 37 °C on day 15. The height of aromagram peaks represents measured odor intensity (percent relative scale). The TIC signal is collected simultaneously, enabling linking odors to specific chemicals. Several unpleasant (out of 20 total) odors were recorded during analysis with GC-MS-O, and few of them were medium intensity.

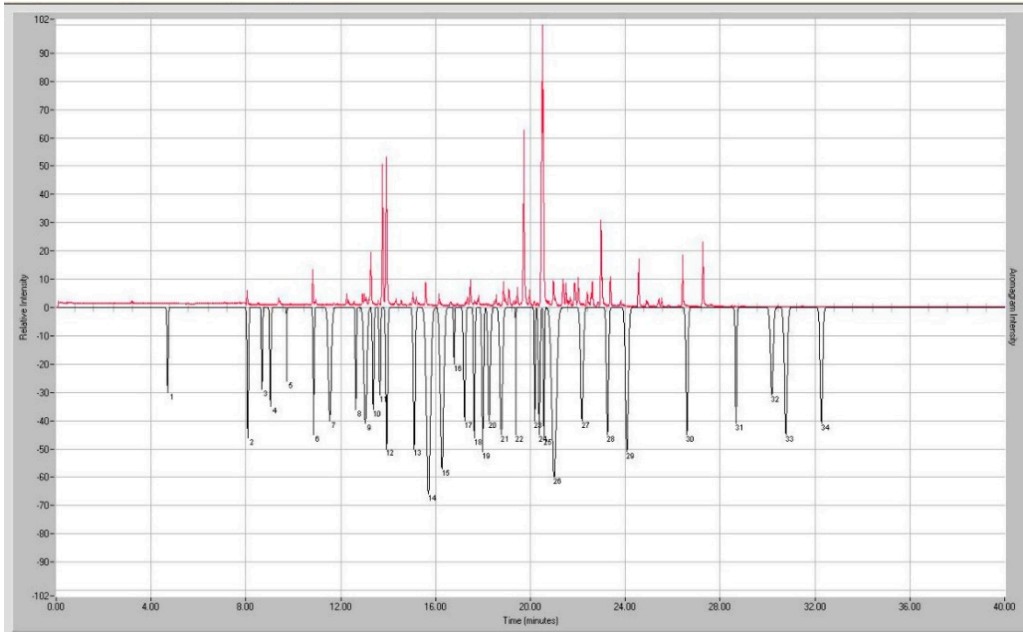

**Figure A3.** An overlay of aromagram (black line) and total ion chromatogram (TIC, red line) of the carpet + subflooring + urine + microbe + Product 3 extracted from headspace by DVB/CAR/PDMS Stableflex (2 cm) SPME fiber for 60 min at 37 °C on day 0. The height of aromagram peaks represents measured odor intensity (percent relative scale). The TIC signal is collected simultaneously, enabling linking odors to specific chemicals. Several unpleasant (out of 34 total) odors were recorded during analysis with GC-MS-O, and few of them were medium intensity.

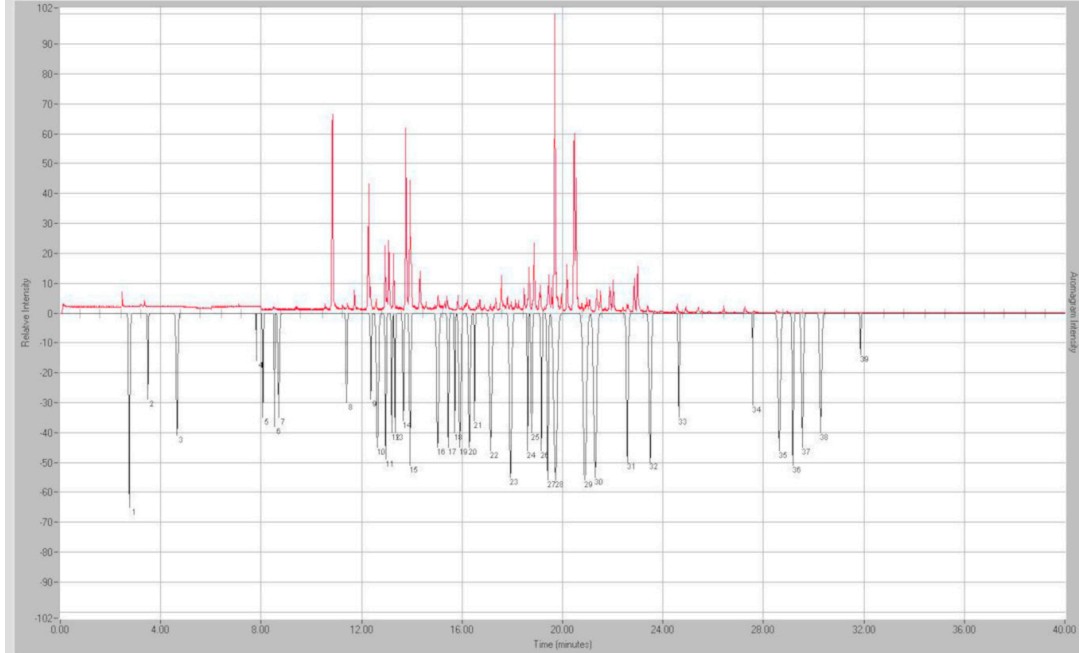

**Figure A4.** An overlay of aromagram (black line) and total ion chromatogram (TIC, red line) of the carpet + subflooring + urine + microbe + Product 3 extracted from headspace by DVB/CAR/PDMS Stableflex (2 cm) SPME fiber for 60 min at 37 °C on day 15. The height of aromagram peaks represents measured odor intensity (percent relative scale). The TIC signal is collected simultaneously, enabling linking odors to specific chemicals. Several unpleasant (out of 39 total) odors were recorded during analysis with GC-MS-O, and few of them were medium-to-strong intensity.

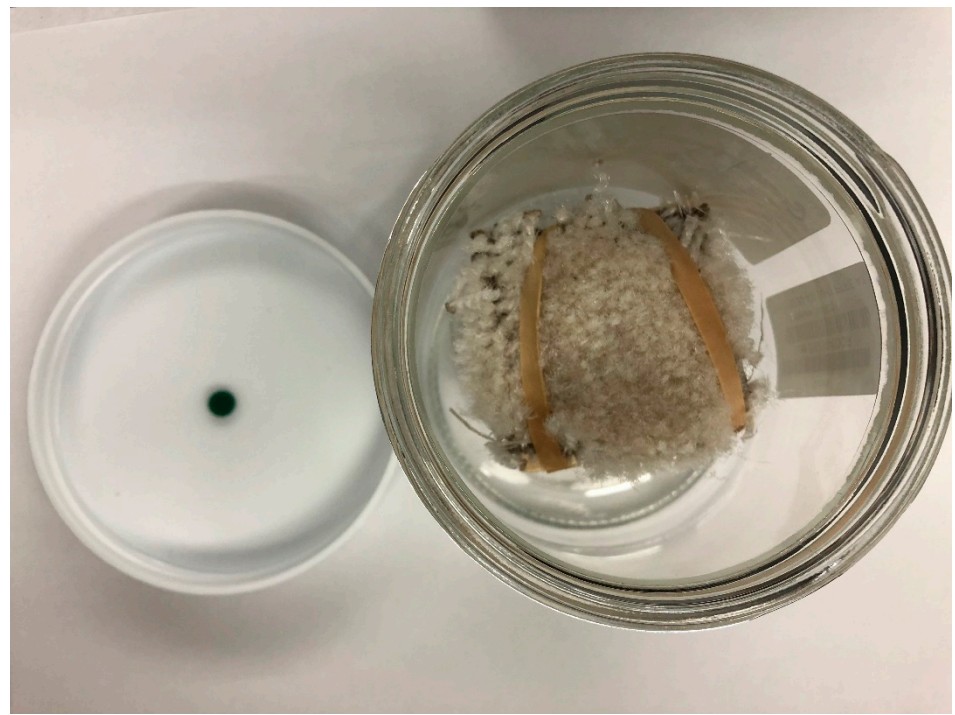

**Figure A5.** The picture shows the test carpet and subflooring specimen with or without treatment inside the 500 mL glass jar. The carpet specimen was tied with the specimen subflooring using a latex rubberband. Thermogreen half-hole septum in the white plastic screw top was used to insert SPME probes inside the headspace. The inside of the white plastic screw top was lined with PTFE.

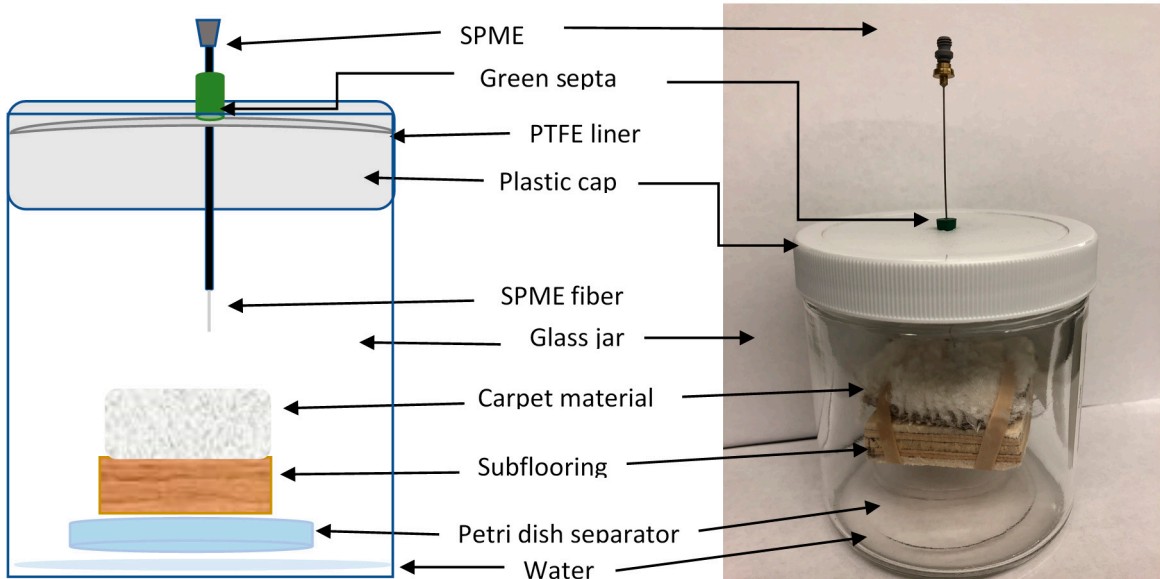

**Figure A6.** The test carpet with or without treatment inside the 500 mL glass jar. Volatiles emitted from carpet with subflooring were sampled by a DVB/CAR/PDMS Stableflex (2 cm) SPME fiber for 60 min at 37 °C. The Petri dish separator was on the bottom of the jar. 10 mL of water was added Day 0 to the bottom of the jar to maintain humidity in the headspace.

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
