# Peer review of "Volatile Compounds Emitted from the Cat Urine Contaminated Carpet before and after Treatment with Marketed Cleaning Products: A Simultaneous Chemical and Sensory Analysis"

_data, 2020_

Round 1

Reviewer 1 Report

Odor nuisance from cat urine is well-known problem and methods to reduce it is needed, so the topic is very relevant. The paper gives relevant data based on a well described methodology.

The ms is a bit wordy and with many repetition. 

Line 18-19: The sentence is incomplete or hanging. Consider to split is up.

Line 20: Micrococcus is be in italic (everywhere)

Line 24: olfactometry -> olfactometer

Line 25: use day 1 (0) and day 15) as in line 18

Line 53-55: Spit up sentence. Don't use "using" twice

Line 62: Add Microsoft first time you mention Excel spreadsheet

Line 71: Table 1, I think. And don't write Table 1 twice.

Line 71-110: Many repetition. Write the general part first, and then add the parts that differ in order to avoid repetition.

Line 215: The overlay of the aromagram and TIC. The black line is TIC and the red line is the odor intensity, I think.
Add a description of odor intensity with a scale. Describe it in the Material section. Is the TIC in Figure A1 from column L in Table S1. Consider to remove peak numbers as it is confusing that the peak numbers don't agree with the numbers in the table.
Why are the peaks after ca. 31 min not listed in the table.

The previous comment goes for Figure A2-A4 as well.

Line 233: Figure A4. What does the solid blue horizontal lines denote?

Reviewer 2 Report

It is an interesting study and data set. Volatile Organic Compounds (VOCs) are one of the major groups of chemical contaminants that poses severe risks in indoor habitants including humans. Although, there are several sources available for indoor contamination of VOCs, flooring carpets (due to its chemical absorbing and slowly releasing nature) are one of the crucial sources, especially if it is contaminated with pet urine. Overall, study design and methodology, analytical characterization and other supporting data are well inscribed.

There are a few comments need to be addressed before considering for publication in Data.

  • Page 2, line 48: The word “his”. Typo?
  • Page 8, line 133: The urine samples were left overnight at the litter box. Since, the target chemicals (VOCs) are highly volatile in nature, how did authors control or validated the actual concentrations? Are there any quality control experiments were performed? If so, please include details.
  • Page 8, line 144: What kind of water was used for the treatment? Since water itself can be a source of introduction of VOCs, that information needs to be provided.
  • Authors can add some of the significant findings at the end of the summary section or user notes. For example, some findings were listed in the abstract but in the main draft.
  • Tables 1 to 4 and Tables S1 to S4: Authors used NIST/Wiley library to obtain the structure or chemical information based on match factors. However, I noticed that the match factors for some of the listed chemicals were poor. So, I strongly recommend adding a denote (*) and explain in table foot note saying “semi-matching or closely matching compounds or compounds identified with less confidence” for anything having matching factor less than 60%.
